# Ultrafast multi-cycle terahertz measurements of the electrical conductivity in strongly excited solids

Z. Chen [1✉], C. B. Curry [1,2], R. Zhang[2], F. Treffert [1,3], N. Stojanovic[4,5], S. Toleikis[4], R. Pan[4], M. Gauthier[1], E. Zapolnova[4], L. E. Seipp[1,6], A. Weinmann[1,6], M. Z. Mo [1], J. B. Kim[1], B. B. L. Witte[1,7], S. Bajt [4,8], S. Usenko[4,9,10], R. Soufli [11], T. Pardini[11], S. Hau-Riege[11], C. Burcklen[11], J. Schein[6], R. Redmer [7], Y. Y. Tsui[2], B. K. Ofori-Okai [1] & S. H. Glenzer [1✉]

Key insights in materials at extreme temperatures and pressures can be gained by accurate measurements that determine the electrical conductivity. Free-electron laser pulses can ionize and excite matter out of equilibrium on femtosecond time scales, modifying the electronic and ionic structures and enhancing electronic scattering properties. The transient evolution of the conductivity manifests the energy coupling from high temperature electrons to low temperature ions. Here we combine accelerator-based, high-brightness multi-cycle terahertz radiation with a single-shot electro-optic sampling technique to probe the evolution of DC electrical conductivity using terahertz transmission measurements on sub-picosecond time scales with a multi-undulator free electron laser. Our results allow the direct determination of the electron-electron and electron-ion scattering frequencies that are the major contributors of the electrical resistivity.

[1] SLAC National Accelerator Laboratory, Menlo Park, CA 94025, USA. [2] University of Alberta, Edmonton T6G-1H7, AB, Canada. [3] Technische Universität Darmstadt, Darmstadt 64289, Germany. [4] Deutsches Elektronen-Synchrotron DESY, Hamburg 22607, Germany. [5] DLR-Institute for Optical Sensor Systems, Berlin 12489, Germany. [6] Universität der Bundeswehr München, Neubiberg 85579, Germany. [7] Institut für Physik, Universität Rostock, Rostock 18059, Germany. [8] The Hamburg Centre for Ultrafast Imaging, Hamburg 22761, Germany. [9] Institut für Experimentalphysik, Universität Hamburg, Hamburg 22761, Germany. [10] European XFEL GmbH, Schenefeld 22869, Germany. [11] Lawrence Livermore National Laboratory, Livermore, CA 94550, USA. ✉email: zchen@slac.stanford.edu; glenzer@slac.stanford.edu

The electrical conductivity of materials shows dramatic variations for different material structures and excitation conditions. For example, under extreme pressure, an insulator's bandgap closes, leading to metallic phases with orders of magnitude increases in the conductivity[1–6]. Some materials exhibit superconductivity with a relatively high transition temperature while under high pressure[7]. At highly elevated temperatures, metals melt and transition into warm dense matter before they turn into plasmas[8,9]. This trajectory is accompanied first by an increased electron scattering frequency in partially Fermi degenerate states reducing the conductivity, the quantum mechanical system then becomes classical[4,10,11], resulting in the inversion of the temperature dependence of the conductivity.

However, precise measurements of the electrical conductivity as materials dynamically evolve through high temperature and high pressure conditions are extremely challenging, because laboratory experiments produce harsh environments that prevent the use of contact electrical probes. While optical and x-ray free electron lasers (FEL) routinely provide a suitable temporal resolution in the femtosecond regime, their utility for determining the conductivity is limited because the information is convoluted by the oscillating fields of the electromagnetic waves, whose frequencies are comparable to or larger than the electron scattering rates. Thus, to extrapolate to the DC electrical conductivity requires a frequency-dependent Drude model or variants[3,4,10,12,13]. Therefore, more complex theories have been applied[14–18] that require experimental testing.

To determine the DC electrical conductivity in excited materials, we have developed the capability to perform conductivity measurements using ultrafast terahertz (THz) radiation that overcomes the high frequency ($>10^{15}$ Hz) electron plasma shielding. The cycles of the THz pulses are sufficiently short to resolve the conductivity changes of transient states on sub-picosecond time scales, and the oscillation period is sufficiently large to allow the carriers to experience DC-like electric fields within their mean-free path. The THz pulses measure the continuous evolution of the electrical conductivity in irreversible excitation conditions in a single event. They resolve the evolution of the electrical conductivity of strongly excited solids at energy densities up to $2 \times 10^{10}$ J/m$^3$, pressures of the order of 10 GPa[9], and ion heating rates that exceed $10^{14}$ K/s producing non-equilibrium superheating conditions[8].

In this study, we measure the THz transmission in FEL heated warm dense gold during the excitation and heating processes with continuous sub-500 fs terahertz pulses. The results allow us to determine the trajectory of the conductivity and total electron momentum scattering frequencies as the material conditions evolve through phase space. This method provides information to discriminate the individual contributing components that underline the electrical conductivity. We find that the electron–electron scattering frequencies shows Fermi-liquid-like behavior and dominates at electron temperatures exceeding $10^4$ K. The electron-ion scattering frequency results support the Ziman theory demonstrating the relation between electrical and structural properties of strongly excited matter. Finally, our results demonstrate that the Drude model can successfully describe the broadband electrical conductivity when interpolating between the THz and optical data, and disclose the shortfalls when extrapolating using only optical data.

## Results and discussion

**Electrical conductivity from THz transmission measurements.** The experiment was performed at Deutsches Elektronen-SYnchrotron (DESY) Free-electron LASer in Hamburg (FLASH) facility, where two sets of undulators provided synchronized extreme-ultraviolet (XUV)-FEL and THz-FEL pulses simultaneously[19–22], cf. Fig. 1. The 150 fs XUV-FEL radiation at a wavelength of $\lambda = 13.6$ nm and with up to 160 µJ pulse energy was used to ionize the $5d$ and $5p$ electrons in 30 nm-thick free-standing gold foils. Subsequently, the sample thermalizes into a degenerate plasma state reaching maximum electron temperatures of $T_e = 16,800$ K, i.e., 1.5 eV at peak intensities of $5 \times 10^{11}$ W/cm$^2$. The multi-cycle THz-FEL pulses containing individual

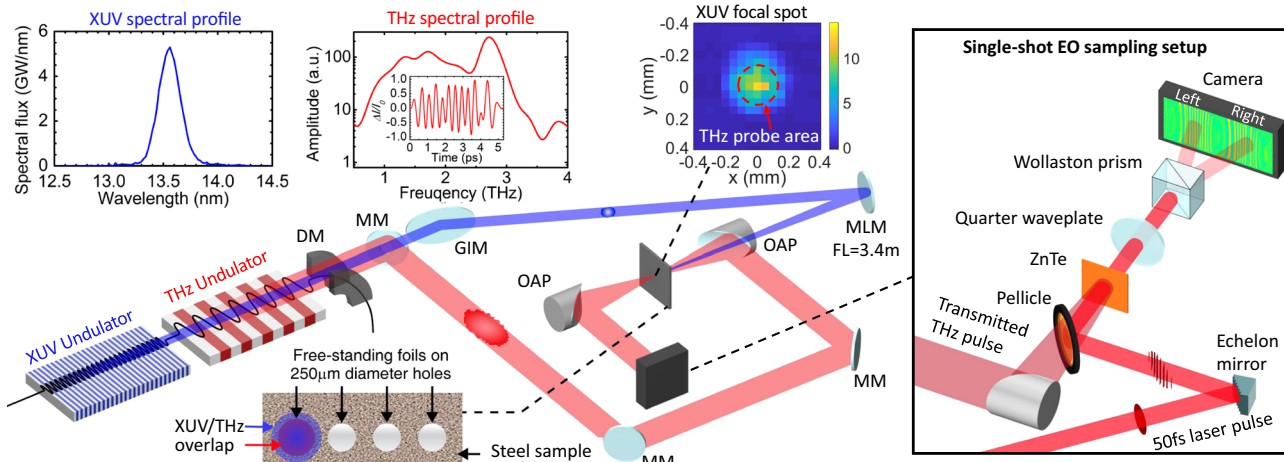

**Fig. 1 The experimental setup to measure terahertz (THz) transmission through free-standing gold foils excited by extreme ultraviolet (XUV) pulses.** The XUV (13.6 nm peak wavelength) and multicycle THz (2.8 THz peak frequency) pulses are generated with two individual undulators using the same electron bunch at ~680 MeV kinetic energy[19,21]. While the residual electrons are bent-off by a dumping magnets (DM), the two photon beams are separated by a metallic mirror (MM) with a through-hole at its center, i.e., the XUV pulses transmit through the hole and the major part of the THz radiation is reflected. After transporting with multiple grazing incidence mirrors (GIM), the XUV pulses are focused onto free-standing gold foils by a near-normal incidence Mo/Si multilayer spherical mirror of 3.4 m focal length. The gold foils are mounted on a steel sample card with 250 µm diameter holes, and the XUV pulses are de-focused to a focal spot of 255 µm × 225 µm FWHM on target. The THz pulses are transported by metallic mirrors before focusing onto the target by a 2″ diameter, $f$/1 off-axis parabolic (OAP) mirror. After transmitting through the sample, the THz pulses are collimated by another OAP, and the $E$-field amplitude is measured by single-shot electro-optic (EO) sampling[23,24] using a 0.5 mm-thick ZnTe crystal as shown in the inset on the right.

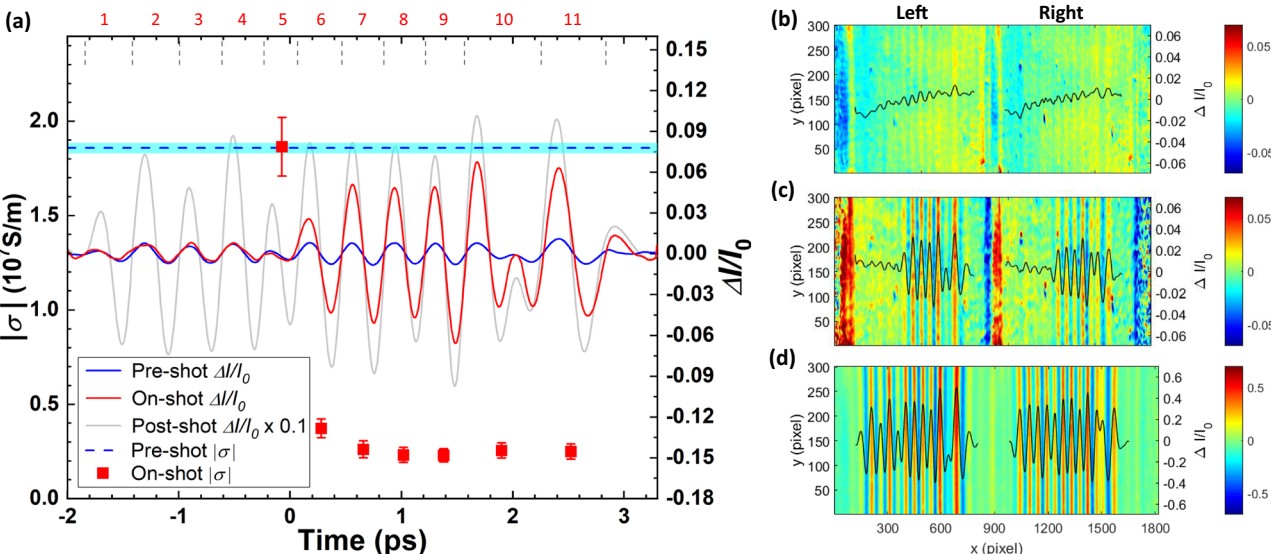

**Fig. 2 Example of single-shot electrical conductivity measurements.** The red squares in **a** are the electrical conductivity σ (left axis) determined from the THz transmission measurements (right axis). $\Delta I/I_0$ is the related intensity change of the electro-optics sampling pulse. The solid red curve in **a** represents the THz transmission through a 30 nm gold film excited with energy densities of 0.91 ± 0.18 MJ/kg, and the blue and gray curves represent the measurements before and after FEL excitation, respectively. The boundary of each THz cycle is marked. The time-resolved electrical conductivity data (σ) are inferred from each cycle, except the first data point which is the average over the five THz cycles that arrive before XUV-FEL excitation. The conductivity of the cold samples measured by pre-shots is illustrated by the blue dashed line with the light blue area indicating the measurement accuracy. In the on-shot measurement, the XUV-FEL heating pulse arrives at 0 ps. No XUV-FEL pulses are present during the pre-shot and post-shot measurements. The error bars in |σ| represent the systematic uncertainty of the measurements. The data in **b–d** show the measured images of single-shot time-resolved THz electric-field profiles[23,24] for the conditions marked by blue, red and gray curves in **a** respectively. Each set of data in **b–d** contains a pair of cross-polarizing images separated by the Wollaston prism, with the corresponding line-outs overlaying on top of the images.

cycles in the range of 0.5–4 THz (see Supplementary Information) with an averaged peak frequency of 2.8 THz ($\lambda = 105$ μm) probe these conditions with a maximum energy of 10 μJ. We apply electro-optics sampling to measure the THz time-domain electric-field waveform, $E(t)$, in a single-shot using a separate 50 fs, 800 nm probe laser pulse. The pulse front of the 800 nm pulse was split by an echelon mirror[23,24], so that distinct spatial areas of the pulse encoded a sequence of time points of the THz waveform. On each shot, we observed a total of 11 THz cycles spanning 5 ps continuously resolving the heating dynamics of the solid with individual cycles shorter than 500 fs.

To determine the electrical conductivity of these samples, we first measure the THz-FEL transmission of the non-excited metallic thin foil (Fig. 2a, b), followed by the transmission through the XUV-FEL heated foil (Fig. 2a, c). The incident THz field is then determined by the transmission through the hole on the sample card after the foil is entirely ablated (Fig. 2a, d). During the pump-probe experiment, we observe a significant increase of the THz transmission almost immediately after the XUV-FEL pulse excites the sample. Before excitation, the time-resolved signal reproduces the THz transmission through a cold foil, which is of order 1%. After excitation, the transmission increases rapidly approaching 10% of the incident field. These significant changes in transmission provide a sensitive and accurate measurement with smaller error bars than THz reflectivity measurements[25]. Because the wavelength $\lambda$ of the THz-FEL is much larger than the sample thickness, we relate the transmission $|t_r|$ and conductivity $\sigma$ in the time domain[26]:

$$|t_r| = \frac{|E_t|}{|E_i|} = \left| \frac{2}{\sigma Z_0 d + 2} \right| \approx \frac{2}{|\sigma| Z_0 d + 2} \quad (1)$$

where $E_t$ and $E_i$ are the transmitted and incident electric-field amplitudes, $Z_0 = 377$ Ω is the free space impedance, and $d = 30$ nm is the sample thickness. Although our THz-FEL pulses

contain a finite bandwidth, at sufficiently low frequency, the complex conductivity $\sigma = \sigma_r + i\sigma_i$ of thin metal films is dominated by the real part that is frequency independent[23,27,28], i.e., $\sigma(\omega) \approx \sigma_r(\omega) \approx |\sigma(\omega)|$, where $\omega$ is the THz angular frequency (see Supplementary Information). Since the electrical conductivity in metals is large, $|\sigma| Z_0 d$ is significantly greater than 2, leading to a $\sigma \sim \frac{1}{t_r}$ scaling. Using Eq. (1), we obtain the electrical conductivity from the amplitude of each THz cycle as shown in Fig. 2a, indicating a rapid reduction of the conductivity right after the XUV-FEL excitation followed by slower gradual changes.

**Comparison of DC electrical conductivity and optical conductivity data.** In Fig. 3, we determine the THz transmission as function of excitation energy density and consequently for various electron temperatures at a time delay of $t = 0.7 \pm 0.2$ ps when the XUV-FEL pulse energy deposition is complete. At this stage, the XUV-FEL excited electrons have recombined, and the electron system has assumed a Fermi distribution[29] characterized by its temperature. The two-temperature model (TTM)[8,30] is applied to calculate $T_e$ and $T_i$ for these conditions (see Supplementary Information). Due to slow electron-phonon coupling[8], at $t = 0.7$ ps the ion temperature ($T_i$) remains small ($T_i \leq 470$ K), and the electron system contains most of the absorbed energy. Figure 3 shows the real part of the electrical conductivity $\sigma_r$ as a function of photon energy from the THz data and from optical data[30] for three different electron temperatures. In ref. [30], the optical conductivity is measured using the same type of sample heated by frequency-doubled Ti:Sapphire laser pulses (400 nm) of 45 fs pulse width. In this case, laser absorption occurs through the excitation of 5d electrons into the conduction band. For this comparison, we choose data at 0.54 ps after laser excitation when the heated electrons are thermalized, resulting in the same temperature and density conditions within the uncertainties as

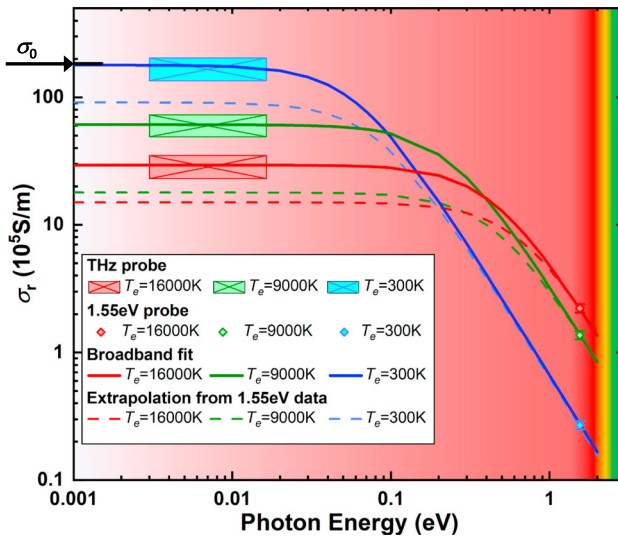

**Fig. 3 The electrical conductivity as a function of photon energy for various excitation conditions of solid gold.** The broadband electrical conductivity data (real part, $\sigma_r$) at electron temperatures of $T_e = 300$ K (blue), $9000 \pm 900$ K (green) and $16,000 \pm 1500$ K (red) are indicated by the different colors; the THz conductivity at small photon energies (0.75–3.75 THz in frequency, or 3–15.5 meV in photon energy) are shown as square blocks that cover the range of frequency components in the THz cycles (horizontal dimension) and uncertainties of the inferred conductivity values (vertical dimension), the optical conductivity at higher photon energies (1.55 eV) from ref. [30] are presented as diamonds, and the DC conductivity at room temperature measured with four-point probe is pointed out by a black arrow. The solid curves represent the Drude model fit through the high and low frequency data, and the dashed curves represent Drude model extrapolation from the optical conductivity values at 1.55 eV using the Drude model. The conductivity measured with THz radiation agrees well with the DC conductivity from four-point probe measurements, while the extrapolation from optical measurements shows significant discrepancies.

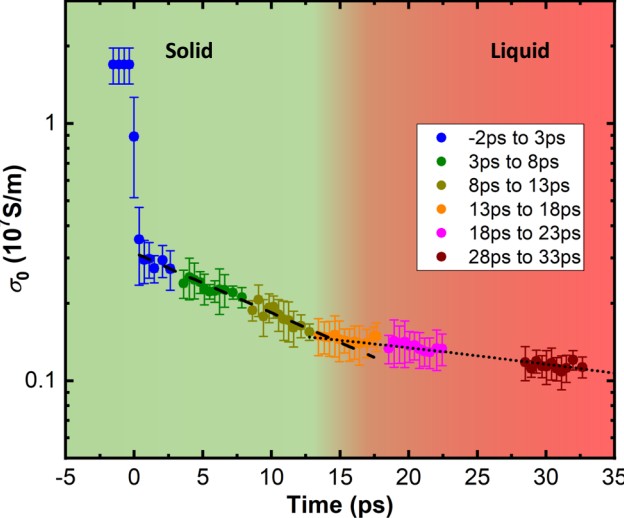

**Fig. 4 Time-resolved electrical conductivity $\sigma_0$ of gold (logarithmic vertical axis) at mean excitation energy density of $0.89 \pm 0.18$ MJ/kg measured by multi-cycle THz-FEL at various time delays.** Each section covers a time interval of 5 ps and the data present an averaged over 4–6 shots. The error-bars include the standard deviation and the systematic error of the measurements. The data after 0 ps are fit by a two-segment exponential decay function represented by the dashed and the dotted lines. The intersection of these two curves at $t = 14.5$ ps coincides with the melting phase transition of gold.

probed by our THz experiments. The optical conductivity measurements average over a target area equal to 1/3 of the full-width-half-maximum (FWHM) of a Gaussian-profile laser pump beam. On the other hand, the present THz experiments probe a slightly larger area equal to the FWHM of the pump profile. Thus, the THz measurements average over a slightly larger range of temperature conditions; they vary less than 12% RMS from the averaged temperatures (see Supplementary Information). Also shown in Fig. 3 is the DC conductivity of our thin foil samples measured by a four-point probe at room temperature. yielding $\sigma_0 = 1.84 \pm 0.15 \times 10^7$ S/m. The latter demonstrates excellent agreement with the conductivity from the THz measurements.

With increasing temperatures, we find that the DC conductivity values decrease, while, on the other hand, the 1.55 eV optical data show the opposite trend. In Fig. 3 we applied the Drude model to relate $\sigma(\omega)$ between the THz[27] and the optical regimes,

$$\sigma(\omega) = \frac{n_e e^2 \tau}{m(1 - i\omega\tau)}, \quad (2)$$

where $n_e$ is the carrier electron density, $e$ is the electron charge, $m = 9.1 \times 10^{-31}$ kg is the electron effective mass in gold[31], $\tau = 1/\nu_e$ is the electron scattering time, which is the inverse of the total electron momentum scattering frequency $\nu_e$. We take the real part of Eq. (2) to fit the data in Fig. 3, i.e., $\sigma_r(\omega) = \frac{n_e e^2 \tau}{m(1 + \omega^2 \tau^2)}$. At $T_e = 300$ K, the electron density results in $n_e = 5.9 \times 10^{28}$ m$^{-3}$, corresponding to one carrier electron per atom[32]. The electron

scattering time is determined using the conductivity data measured in three regimes by 1) the four-point probe at zero frequency, 2) the THz-FEL probe at 2.8 THz peak frequency, and 3) the optical probe laser at 1.55 eV. At elevated temperatures, only the THz and optical conductivity data are used to solve for $n_e$ and $\tau$ in the Drude formula. Figure 3 shows that the Drude model can satisfactorily describe the frequency dependency of the electrical conductivity from the AC to the DC regime if data are available in both regimes. Further, we find that the frequency response of $\sigma_r(\omega)$ is effectively constant for photon energies from zero to 0.012 eV, resulting in $\sigma_r \approx \sigma_0$ in this regime, i.e., the DC conductivity value is identical to the THz conductivity. The dashed curves in Fig. 3 indicate the results when the same model is applied using only the optical data at 1.55 eV to extrapolate to the DC conductivity regime. It leads to significantly lower DC conductivities $\sigma_0$ than obtained by our THz measurements, demonstrating the need for measurements in the THz regime (also see detailed comparisons in the Supplemental Information).

**Temporal evolution of the electrical conductivity**. By varing the delay between the XUV-FEL pump and the THz probe pulse train allows us to measure the temporal evolution of the electrical conductivity over a 35 ps-long time window, spanning the transition from the solid state to a laser-excited solid and into the warm dense matter (plasma) state. Figure 4 shows the electrical conductivity as a function of time for an energy density of $0.89 \pm 0.18$ MJ/kg and assuming constant sample thickness for the whole time interval. Following the abrupt decrease right after $t = 0$, the conductivity continues to decline at a slower rate. It follows an exponential trend until about $t = 15$ ps as indicated by the linear dependence on the semi-log plot, cf. Fig. 4. However, at $t = 14.5$ ps we observe a discontinuity determined by a two-segment exponential fit; this time coincides with the melting phase transition and the onset of hydrodynamic expansion[8,9]. The sample expansion at $t > 15$ ps is likely accompanied by an increased sample thickness $d$ as well as the development of density

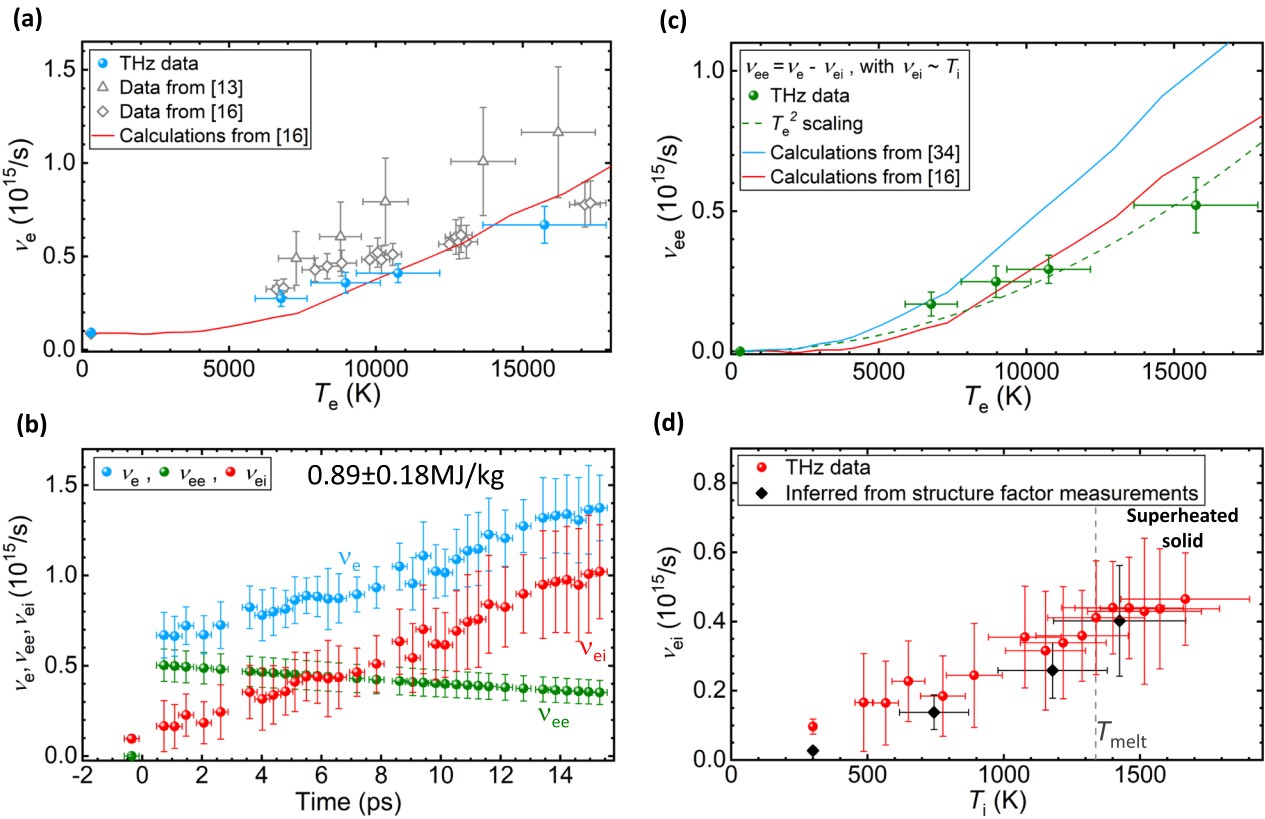

**Fig. 5 Electron scattering frequencies as functions of time and temperatures determined from broadband conductivity fit using the Drude model. a** shows the total electron scattering frequency ($\nu_e$) as a function of electron temperature $T_e$ determined from this work (blue symbols), compared to the data reported from optical measurements[13,16] (gray open triangles and diamonds) and the theoretical calculations from ref. [16] (red curve). **b** $\nu_e$ determined from the temporally resolved THz conductivity measurements (blue circles), which are the sum of the electron-electron scattering $\nu_{ee}$ (green circles) and the electron-ion scattering frequencies $\nu_{ei}$ (red), and at negative time delay the data for $\nu_e$ and $\nu_{ei}$ overlap. **c** $\nu_{ee}$ data obtained from the total electron scattering frequency $\nu_e$ (green symbols), and the green dashed curve is a fit to the data with a $T_e^2$ scaling. The blue and red curves are the theoretical calculations from refs. [16,34]. **d** $\nu_{ei}$ data as a function of ion temperature $T_i$ (red symbols), and the black diamonds represent $\nu_{ei}$ calculated from the ion-ion structure factor ($S_{ii}$) measured in the ref. [8] utilizing the Ziman formula[35], indicating excellent agreement between these two methods. The horizontal error bars in **a**, **c**, and **d** have taken into account the FEL energy fluctuation and temperature distribution over the THz probed area, and in **b** they represent the width of the THz cycles. The vertical error bars are determined by the precision of the measured electrical conductivity, and the error bars of the electron temperatures take into account both the uncertainties of the experimental measurements and the range of temperatures as determined by the TTM calculations.

gradients. Future work will combine these data with independent measurements of the density profile to determine the transition to the classical plasma regime. These observations show that the multicycle THz probe determines $\sigma_0$ accurately for strongly excited solids and provides information on phase transitions. In the following section, we focus on the data up to $t = 15$ ps since the target thickness is well constrained until melting and, in addition, independently measured accurate structure factors exist for these conditions.

**Time and temperature dependence of electron scattering frequencies.** Applying the Drude model (Fig. 3) to the time-dependent THz data determines the total electron scattering frequency $\nu_e$. The data show a rapid increase right after the XUV-FEL excitation, while the free carrier density $n_e$ only increases slightly, i.e., by a factor of 1.3. The $\nu_e$ data as function of $T_e$ are shown in Fig. 5a. At the maximum electron temperature of $T_e = 16,000 \pm 1000$ K, we find an increase of $\nu_e$ over six times compared to the room temperature value. The significant increase of $\nu_e$ explains the opposite temperature dependence of $\sigma_r$ observed between the THz and the optical regimes in Fig. 3, with $\omega/\nu_e = \omega\tau \ll 1$ in the THz regime results in $\sigma_r \propto \frac{1}{\nu_e}$ according to Eq. (2),

and $\omega/\nu_e \gg 1$ in the optical regime leads to $\sigma_r \propto \nu_e$. The $\nu_e$ data extrapolated from the optical conductivity data using the original Drude model[13] and a modified Drude model[16] show even higher values than the THz data, *cf.* Fig. 5a. The significantly higher $\nu_e$ data reported in ref. [13] can be explained by the following: (1) this study uses 800 nm, 30 fs laser pulses to excite 300 nm-thick samples, which could lead to larger longitudinal energy density gradients at the time of measurement at 0.1 ps after laser excitation; (2) the use of fairly thick targets prevents the use of TTM to determine the temperature and estimation of $T_e$ using electron density of states are complicated by effects of electron orbital hybridization[16]. In comparison, the theoretical calculations reported in[16] agree well with our measurements at $T_e < 10,000$ K but slightly overestimate the experimental results at higher temperatures.

In Fig. 5b, we show the temporal evolution of $\nu_e$ at an absorbed energy density of $0.89 \pm 0.18$ MJ/kg, where we see an initial jump of $\nu_e$ right after the excitation at $t = 0$ ps, and a subsequent slow increase at later time delays. The former behavior is synchronized with the instantaneous heating of electrons, when little changes have occurred for the ion temperature. Thus, we ascribe this jump to a rapid increase in $\nu_{ee}$. Consequently, we can determine $\nu_{ee}$ as a function of $T_e$ as shown in Fig. 5c, $\nu_{ee}(T_e) = \nu_e(T_e, T_i) - \nu_{ei}(T_i)$,

using the $\nu_e$ data presented in Fig. 5a. Here we approximate $\nu_e \approx \nu_{ei}$ at room temperature and $\nu_{ei} \propto T_i$ at $T_i$ near 300 K[32].

In Fig. 5c, we find the $\nu_{ee}$ data follow a $T_e^2$ scaling indicating Fermi-liquid behavior with $\nu_{ee} = A \frac{1}{\hbar} \frac{(k_B T_e)^2}{\varepsilon_F}$, where $\hbar$ is the reduced Planck constant, $k_B$ is the Boltzmann constant, and $\varepsilon_F = 5.5$ eV is the Fermi energy of gold. We find the scaling factor $A = 0.95$ well agrees with the prediction from the ref. [32]. Quantifying $\nu_{ee}$ in strongly excited solids has been controversial[13,14,17,33,34] and has hitherto not been possible to determine from experimental data. More importantly, different than in metallic hydrogen, where a transition from a Fermi-degenerate metal to a Maxwellian plasma has been observed for temperatures of of 0.15–0.4 times the Fermi temperature ($T_F$) in ref. [4], we do not observe the breakdown of Fermi-liquid behavior in gold for $T_e$ up to $1/4 T_F$. The separation of electron–ion and electron–electron scattering components offers a first direct test of the theory, and allows further exploration of the boundary between the quantum mechanical and the classical behavior in metals. Also shown in Fig. 5c are the theoretical calculation results of $T_e$ dependent $\nu_{ee}$ values[16,34]. The calculations from ref. [34] overestimate the contribution of $\nu_{ee}$ scattering processes to the total electrical conductivity. The data from ref. [16] is a refined calculation of ref. [34] that renders much better agreement with our experimental results at $T_e \leq 10{,}000$ K. In ref. [16] only the scattering between $6s$ and $5d$ electrons is considered, and the scattering between $6s$ electrons is ignored because only a small portion of it can contribute to the electrical conductivity by the Umklapp process[32]. The discrepancy at $T_e > 10{,}000$ K is consistent with the comparison of $\nu_e$ in Fig. 5a, suggesting the need for further improvement of the theory.

Based on the $\nu_{ee}$ data as function of $T_e$, we plot the time-dependent $\nu_{ee}$ in Fig. 5b using the $T_e$ obtained from the TTM calculations. Subsequently, we obtain $\nu_{ei} = \nu_e - \nu_{ee}$ at each time step as the samples enter the superheating regime[8]. The complex interplay between $\nu_{ee}$ and $\nu_{ei}$ is revealed: (1) immediately after XUV-FEL excitation, $T_e$ reaches $16{,}000 \pm 1000$ K and $\nu_{ee}$ greatly exceeds $\nu_{ei}$; (2) the energy coupling from the electrons to the ions results in the rise of $\nu_{ei}$ accompanied by the decline of $\nu_{ee}$; (3) near the completely molten state at ~15 ps, $\nu_{ei} \approx 3\nu_{ee}$ when $T_i$ is merely a quarter of $T_e$. Such observations indicate that $\nu_{ee}$ makes a substantial contribution to the electrical conductivity at $T_e \geq 10{,}000$ K, however $\nu_{ei} \gg \nu_{ee}$ under equilibrium conditions.

In Fig. 4d, we plot the electron-ion scattering frequency $\nu_{ei}$ as a function of ion temperature $T_i$. We find that the $\nu_{ei} \propto T_i$ scaling still holds up to the melting temperature $T_{melt} = 1337$ K. The theoretical calculations of $\nu_{ei}$ for high energy density matters are usually performed using the Ziman theory[17,33,35] that requires the knowledge of the ion–ion structure factor ($S_{ii}$). However the correct modeling of $Sii$ in this regime is very challenging due to strongly coupled ions and complex electron screening[15,36]. Therefore, we use the experimentally measured ion-ion structure factor ($S_{ii}$) from ref. [8] to calculate the $\nu_{ei}$ applying the Ziman formula[35,37] (see Supplemental Information). We find close agreement of the $\nu_{ei}$ data determined from these different approaches, demonstrating the connection between electronic and ionic structural properties in strongly excited systems.

In conclusion, using multicycle THz radiation, we demonstrate a sensitive, time-resolved measurement of the electrical conductivity of intense ultrafast free-electron laser excited gold. We have shown that in the warm dense matter regime the DC conductivity is well approximated by the electrical conductivity at THz frequencies. Our measurements provide high-quality data that test existing theoretical calculations and reveal the limitation of extrapolating optical conductivity measurements to the DC regime. This type of experiment allows us to measure the

electrical conductivity on ultrafast time scales enabling the simultaneous determination of the electron scattering frequency as function of time and temperature, and identify the individual contributions from electron–electron and electron–ion scattering. This work further points to an avenue to understand the relation of the material structure and the electrical conductivity at strongly excited irreversible conditions, inspiring applications of THz radiation for the study of high-intensity laser-matter interactions, material science, and the research of planetary interiors. We also expect that this type of studies can be performed in the future with table-top laser systems, i.e., heating the samples by high intensity ultrafast laser pulses and probing them by high brightness laser-generated single-cycle[38] or multi-cycle THz radiation[39].

## Methods

**XUV-FEL pump pulses**. Ultrafast heating pulses with a wavelength $\lambda = 13.6$ nm are produced using an XUV-FEL operating in the self-amplified spontaneous emission (SASE) beam mode. The kinetic energy of the electrons is 680 MeV and the radiation is produced using a 12 mm gap undulator. The XUV pulse energy is measured by gas monitor detectors on each shot, allowing us to determine the energy density of each measurement. The pulse width of ~150 fs FWHM is estimated based on the measured electron bunch length using the transversely deflecting RF-structure. The XUV pulses are transported to the FLASH BL3 experimental area with four grazing incidence mirrors. The total efficiency of these mirrors is estimated to be 70%. The XUV pulse is then focused by a Mo/Si multi-layer coated super-polished spherical mirror, whose nominal radius of curvature is 6,850 mm and a peak reflectivity of 45% at 13.6 nm. The samples are placed before focal spot of the mirror so that a larger area on sample can be heated. The XUV spot size on target is measured by spatially scanning a pyro-detector with a 50 μm diameter pin-hole mounted on its front surface, and its energy distribution is fitted by Gaussian profiles indicating vertical and horizontal FWHM of 255 μm and 225 μm, respectively. Half of the FEL energy is enclosed in the FWHM area, resulting in the maximum intensity of $5 \times 10^{11} W/cm^2$ on target. About 77% of the incident XUV energy is deposited in the 30 nm thick gold foils[40], and the energy density gradient along the longitudinal direction is expected to smooth out within 500 fs after the excitation by ballistic electron energy transport and thermal diffusion (see Supplementary Information).

**THz-FEL probe pulses**. After generating the XUV pulse, the same electron bunch passes through a planar electromagnetic undulator to generate the multicycle THz pulses[19]. The electromagnets are tuned to produce 2.8 THz emission (determined from the complete THz pulse train) in our experiment. The averaged frequencies of the individual THz cycles are in the range of 2–3 THz (see Supplementary Information). Some additional THz radiation is also emitted when the electrons are deflected by the bending magnet. The THz radiation is transported to the target chamber by a few metallic mirrors[19,21]. Inside the target chamber, the THz pulse is focused on the target surface by a 2″ diameter, 2″ effective focal length (EFL) 90° off-axis parabolic mirror (OAP). The transmitted beam is collimated by an identical OAP. Both of these OAPs have 3 mm holes at the center to allow the XUV pulse to be collinear with the THz pulse on the sample. The THz pulse is then focused by a third OAP of 2″ diameter and 3″ EFL onto a 0.5 mm thick ZnTe (110) crystal.

The temporal profile of the THz electric field (E-field) is measured by a single-shot electro-optic (EO) sampling setup with balanced configuration as illustrated on the right hand side of Fig. 1 in the main text. Here, a time-synchronized linearly polarized 800 nm, 50 fs laser pulse is reflected from a stepped echelon mirror to create a tilted wave front to cover a 6 ps time window[23,24]. The echelon mirror contains 120 steps of 7.5 μm in height and 150 μm in width. A pellicle beam splitter is used to overlap the THz and 800 nm pulses, and both are focused onto the ZnTe crystal (focusing optics for the laser is not shown). The THz E-field causes a transient birefringence in the ZnTe crystal via the Pockels effect, and this birefringence causes a polarization change in the 800 nm pulse. The resulting polarization change is measured by a camera located after a quarter waveplate and a Wollaston prism[23,24,41]. The quarter waveplate further modifies the polarization and the Wollaston prism separates the beam into two arms with orthogonal linear polarizations (e.g., horizontal and vertical). The two beams are imaged simultaneously onto the same CMOS camera (imaging optics not shown). On each image, the temporal information of the THz electric field is encoded along the horizontal direction due to the tilted pulse front. The relative amplitude of the THz E-field can then be quantified by comparing the change in brightness ($\Delta I$) at different time delays due to the THz-induced EO effect with the original brightness ($I_0$) without THz. Because the E-field induces opposite brightness changes on the orthogonal polarized images, the signal to noise ratio is enhanced by subtracting the signal of one arm from the other, i.e., $\frac{\Delta I}{I_0} = \frac{\Delta I^R}{I_0^R} - \frac{\Delta I^L}{I_0^L}$[24,41], where $R$ and $L$ denote the images on the right and left side of the camera respectively, separated by the

Wollaston prism. Examples of the acquired data using this balanced detection method are shown in the Supplemental Information.

**Sample preparation and characterization.** The 30 nm-thick polycrystalline gold foils are prepared by e-beam evaporation coating on a polished sodium chloride (NaCl) surface. The thickness of the samples is controlled by a quartz crystal monitor during the deposition. The NaCl substrates are subsequently dissolved in de-ionized water to release the gold thin foils onto the surface of the water. The gold foils are lifted off and mounted on the 1 mm-thick steel sample cards with arrays of 90° cone-shaped holes of 250 μm smaller diameter separated by 2.5 mm. These holes efficiently limit the area probed by the THz-FEL to be the same as the area heated by the center of the XUV-FEL.

Some of the gold foils are placed on glass slides to measure their thicknesses and DC electrical conductivity. For the thickness measurement, we use an atomic force microscope to scan the step height across the thin film-glass boundaries, and a sample thickness of $30 \pm 2$ nm is found. The DC electrical conductivity is measured by a four-point probe instrument, yielding $\sigma_0 = 1.84 \pm 0.15 \times 10^7$ S/m.

**Temporal and spatial overlap of the XUV and THz pulses.** Proper spatial and temporal overlap between the XUV and THz pulses on target is an important component of this experiment. The spatial overlap is achieved by first maximizing the THz transmission through an open hole on the sample card. We then maximize the XUV pulses through the same hole. A thin Ce:YAG scintillator located 40 cm after the sample card is used to monitor the transmitted XUV signal. The temporal overlap is found by searching the onset of THz transmission through a 500 μm-thick silicon wafer right after it is heated by the XUV pulse.

**Data acquisition.** In our measurements, we take three sets of data on each gold sample: (1) THz transmission through the cold sample (pre-shot); (2) THz transmission through the XUV heated sample (on-shot); (3) THz transmission through the same hole after the sample is completely ablated by the XUV pulse (post-shot). For each thin film, only one on-shot measurement is taken, while 30 pre-shot and 10 post-shot data to improve the signal-to-noise ratio for the other two measurements.

## Data availability
The authors declare that the data supporting the findings of this study are available within the paper and its Supplementary Information file. All other relevant data supporting the findings of this study are available on request.

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

## Acknowledgements
This work was supported by DOE Office of Science, Fusion Energy Science under FWP 100182. It was also supported by the Natural Sciences and Engineering Research Council (NSERC) of Canada. This work was performed under the auspices of the U.S. Department of Energy by Lawrence Livermore National Laboratory under Contract No. DE-AC52-07NA27344. Z.C. acknowledges the Stephenson Distinguished Visitor Programme from DESY Photon Science. C.B.C. acknowledges partial support by the NSERC, and F.T. acknowledges the support from the National Nuclear Security Administration (NNSA). N.S., R.P., and E.Z. acknowledge the support from German Academic Exchange Service (DAAD Grant Numbers 57393513 and 57514761). M.Z.M. acknowledges partial support from the U.S. Department of Energy, Laboratory Directed Research and Development

(LDRD) program at SLAC National Accelerator Laboratory, under contract DE-AC02-76SF00515. S.B. acknowledges support of the Cluster of Excellence' CUI: Advanced Imaging of Matter' of the Deutsche Forschungsgemeinschaft (DFG)—EXC 2056—project ID 390715994. The gold thin film samples were prepared at the Stanford Nano Shared Facilities (SNSF), supported by the National Science Foundation under award ECCS-1542152. The authors also wish to thank Klaus Sokolowski-Tinten from the University of Duisburg-Essen for providing the experimental chamber.

## Author contributions

Z.C., J.S., Y.Y.T., B.K.O.-O., and S.H.G. conceived the idea of this study and planned the experiments. Z.C., C.B.C., J.B.K., and B.K.O.-O. designed the experiments. Z.C., C.B.C., R.Z., F.T., N.S., S.T., R.P., M.G., L.E.S., A.W., S.U., and Y.Y.T. carried out the experiments. N.S., S.T., R.P., and E.Z., set up and optimized the FELs for the experiments. Z.C., M.Z.M., B.K.O.-O., and S.H.G. analyzed and interpreted the data. R.S., T.P., S.H.-R., and C.B. designed, fabricated, and characterized the multilayer XUV mirror. R.Z. characterized the samples. J.B.K. and S.B. designed and improved the sample cards. B.B.L.W. and R.R. performed the simulations for this study. Z.C., M.Z.M., B.K.O.-O., and S.H.G. wrote the manuscript with inputs from all authors.

## Competing interests

The authors declare no competing interests.
