## [Peer Review File · Nature Communications]

Reviewers' Comments:

Reviewer #1:

Remarks to the Author:

This article described an experiment performed at FLASH in Hambourg designed to measure electrical conductivity for gold following a ultrafast laser excitation. Conductivity is deduced from the measured change in transmission.

Compared to previous experiment as the one described in Ref. 16 for example, this experiment leads almost directly to DC conductivity (assuming that it is constant on 15 meV which is a reasonable assessment).

DC conductivity is clearly needed for WDM community and from that point this experiment is a step forward. This paper deserves to be published in Nature Communications.

One interesting result here, is the difference they have on the extrapolated DC value with the value extrapolated from the 1.55eV measurement.

They found the same behavior with temperature but they obtain higher value for DC conductivity. This shows the importance of the present measurement compared to previous experiment to constrain the models. But a discussion is missing here on the possible impact of the experimental set up on the measured (deduced) quantities.

On the same line, the authors compare their results with ref. 13 and 16. But there is no discussion on the impact of the experimental set up on deduced quantities. For example in ref 13, the gold sample is 300 nm thick and the pump laser is 800nm 30fs. and in ref 16, pump laser is 400nm and 45fs. The authors should discuss in more detail the comparison with published experimental results.

It seems strange to me that the author choose to present time evolution of electrical conductivity in the supplemental (Figure S6) as this is an important result for the community. It should be part of the paper. Time evolution of DC conductivity allows for comparison with time dependent models. For example, this is a way to test electron-phonon coupling which is another problematic quantity for WDM.

I am less convinced by the theoretical interpretation of the results.

I don't understand why the authors introduce a Drude-Smith model? Following ref. 27, this formulation could be necessary near the percolation regime i.e. for film thickness around 7nm. With a 30 nm film, there is no need to introduce this model. At the end, they determine a C value - 0.17 ± 0.16 . Considering the error bars, the C value is nearly 0 as expected from 27 and the two curves (Drude and Drude-Smith are identical).

The author should justify why they think that they need this Drude-Smith model for their experimental condition. In my opinion, the use of Drude-Smith model here is not relevant. There is a typo in the reference for Drude-Smith model, it should be PRB 64 155106 (2001).

Concerning electron scattering frequency, they use simple model and many assumptions to deduce ω_{ee} and ω_{ei} scattering frequencies and I am not sure that the conclusions drawn are precise enough to write at the end of the abstract that they "elucidate the microscopic origin of electrical resistivity".

Reviewer #2:

Remarks to the Author:

Dear Editor,

I have reviewed the paper by Z. Chen " Ultrafast Multi-cycle Terahertz measurements of the electrical conductivity in strongly excited solids". The reported experiment is using a pump-probe approach: a 13.5 nm pump at intensity $>2 \text{ TW/cm}^2$ and a 100-300 nm relatively broadband probe to measure

time dynamics of ionization and heating/melting of a golden nanofoil. Because the wavelength of the probe is much larger than the sample thickness, they relate the measured amplitude of the transmitted probe first to foil conductivity (before the pump pulse arrived) and then to the plasma conductivity. The former is calibrated against a true direct measurements of DC conductivity of the foil on the absolute scale. The deduced plasma conductivity is correlated to the initial foil conductivity. Overall, the experimental techniques, especially in part of time resolved single-shot electro-optic sampling for THz analysis, are impressive. The results obtained are significant and interesting for the laser-fusion and high-energy density physics communities. Combination of novel parameter space (relatively cold ~ 1 eV XUV driven plasma) and the results qualify this paper for publication in Nature Communications. However, several specific issues presented below are ought to be addressed prior to publication.

1) Throughout the paper the authors make impression that they present direct measurements of DC electrical conductivity. Without any doubt data on THz transmission correlate with electrical conductivity, as was shown e.g. in Ref. 25 for THz probe reflection. But it needs to be clear stated that these are not direct measurements of DC electrical conductivity.

2) In several places the authors call the ionization by 13.5 nm radiation as XUV excitation. Typically excitation process is the process that takes place below the ionization threshold and quantum structure of atoms plays a role. Peak Intensities of the pump in the study around $2\text{TW}/\text{cm}^2$ are much above the typical ablation threshold of GW/cm^2 , so plasma is generated. Indeed the plasma is non-Maxwellian and far from equilibrium but it is still a plasma and this is ought to be clarified.

3) Following 2). It will be useful for a reader to know that T_e is ~ 1 eV and the XUV intensity $\sim 2\text{TW}/\text{cm}^2$ to be conveniently connected with other laser-plasma studies.

4) At the very core of the paper is comparison of measured and deduced from measurements electrical conductivity of Au with multiple theoretical models including a common Drude, Drude-Smith, etc models. But there is a lack of comparison with data obtained by other groups, e.g. from Ref. 25 on Al nanofoils, etc.

5) The authors are using an XUV FEL laser operating in the SASE regime which is famous for a random spike structure in the time domain. These are rather difficult to account for. How high intensity spikes may influence their measurements and how it was included in analysis of the total experimental uncertainty?

6) In Figure 2a, transmission on-shot and post-shot are presented on the same time scale from -2 to 3 ps. It is confusing to see postshot data before the shot arrived. Please elaborate. Also it will be useful to make an inset with a zoom-in image of a raw data. Now it is very difficult to digest.

Reviewer #3:

Remarks to the Author:

Overall the article is well written and likely to be of interest to different communities.

One issue is with the use of the Drude-Smith model over the Drude model. Based on the fitting the authors obtain a value for C of -0.17 ± 0.16 implying that the Drude model only sufficiently fits the data based on the error. While an alternative value of C of -0.17 may be a better fit, this also complicates the model. Additionally this error bound represents a relatively large portion of the allowed values for C . The authors should comment on the error bars and ways to reduce the error in the future either with more detailed measurements, using data from other experiments, etc.

In figure 4, the authors should add the fits for the Drude model in addition to the Drude-Smith model to better match figure 3 and allow the reader to better understand the differences. Adding this

information to 4a should be sufficient if adding it to b-d makes the figure difficult to read. The authors should clarify the sentence (bottom of page 3) "Finally, our results demonstrate the shortfalls of a simple Drude extrapolation." to clarify extrapolation from optical frequencies. The authors should briefly mention the accuracy and error range of the two temperature model in the text.

Point-by-point response to reviewers' comments

Reviewer #1 (Remarks to the Author):

This article described an experiment performed at FLASH in Hamburg designed to measure electrical conductivity for gold following a ultrafast laser excitation. Conductivity is deduced from the measured change in transmission.

Compared to previous experiment as the one described in Ref. 16 for example, this experiment leads almost directly to DC conductivity (assuming that it is constant on 15 meV which is a reasonable assessment).

DC conductivity is clearly needed for WDM community and from that point this experiment is a step forward. This paper deserves to be published in Nature Communications.

We appreciate the comments by Reviewer #1. We have complied with all requests and answered all questions and comments in the detailed point-by-point responses below. Thank you for your time and for your assessment.

One interesting result here, is the difference they have on the extrapolated DC value with the value extrapolated from the 1.55eV measurement.

They found the same behavior with temperature, but they obtain higher value for DC conductivity. This shows the importance of the present measurement compared to previous experiment to constrain the models. But a discussion is missing here on the possible impact of the experimental set up on the measured (deduced) quantities.

Response to the above comment:

Thank you for suggesting to us to include a discussion related to the experimental set up and the impact on the measured quantities. We comply with the suggestion and provide the discussions of our experimental setup, including both the XUV-FEL heating and the THz probing, and their potential influence on our measurements.

Although the initial XUV-FEL absorption and heating mechanisms are different compared to optical lasers, however, at the time when we probe the conductivity ($t = 0.7$ ps) the electron system has thermalized and assumed a Fermi distribution [N. Medvedev et al., Phys. Rev. Lett. 107, 165003(2011)]. The temperatures are the same within the estimated uncertainties, and a comparison between XUV-FEL and optical laser experiments can be justified.

To address this point, we have made two changes in our manuscript:

(1) In the first paragraph of the section “Electrical conductivity from THz transmission measurements”. The following sentences are changed from

The 150 fs XUV-FEL radiation at a wavelength of $\lambda=13.6$ nm with up to 160 μ J pulse energy was used to excite 30 nm-thick free-standing gold foils to a maximum electron temperature of $T_e=16,000$ K ...

to

The 150 fs XUV-FEL radiation at a wavelength of $\lambda=13.6$ nm and with up to 160 μJ pulse energy was used to ionize the 5d and 5p electrons in 30 nm-thick free-standing gold foils with a peak intensity near $5\times 10^{11}\text{W}/\text{cm}^2$. Subsequently, the sample thermalizes into a degenerate plasma state reaching maximum electron temperatures of $T_e=16,000$ K ...

(2) In the first paragraph in the section of “Comparison of DC electrical conductivity and optical conductivity data”, we have added the following sentence:

At this stage, the XUV-FEL excited electrons have recombined, and the electron system assumed a Fermi distribution (29) characterized by its temperature.

In addition, our THz probe pulses irradiate a target area equal to the full-width-half-maximum (FWHM) of the XUV-FEL pulse. This area is slightly larger than that chosen in optical conductivity measurements, i.e. in Ref. 16 the conductivity data are determined from 1/3 of the FWHM target area. Therefore, the THz measurement averages over a slightly larger range of temperatures in the samples. This range is found to be 12% RMS.

To discuss this point, we have made the following changes in the manuscript:

(1) In the first paragraph in the section of “Comparison of DC electrical conductivity and optical conductivity data”, we have added the following sentence:

The optical conductivity measurements average over a target area equal to 1/3 of the full-width-half-maximum (FWHM) of a Gaussian-profile laser pump beam. On the other hand, the present THz experiments probe a slightly larger area equal to the FWHM of the pump profile. Thus, the THz measurements average over a slightly larger range of temperature conditions; they are found to be within 12% RMS of the averaged temperatures (see Supplementary Information).

(2) In Fig. 5 (a) and (c) (originally Fig. 4(a) and (c)), the temperature error bar includes the RMS temperature distribution in the THz probed area (please see Fig. 5 below). In the figure caption, the following sentence is added:

and the error bars of the electron temperatures take into account both the uncertainties of the experimental measurements and the range of temperatures as determined by the TTM calculations.

Fig. 5 (originally Figure 4 in the manuscript): **Electron scattering frequencies as functions of time and temperatures determined from a broadband conductivity fit using the Drude model.**

On the same line, the authors compare their results with ref. 13 and 16. But there is no discussion on the impact of the experimental set up on deduced quantities. For example in ref 13, the gold sample is 300 nm thick and the pump laser is 800nm 30fs. and in ref 16, pump laser is 400nm and 45fs. The authors should discuss in more detail the comparison with published experimental results.

Response to the above comment:

We comply with this request and in the new version of the manuscript we have added a brief discussion of the targets and laser conditions reported in Refs. 13 and 16.

In the first paragraph of the section “Comparison of DC electrical conductivity and optical conductivity data” we added a brief discussion about the experiments reported in Ref. 16:

In Ref. 16, the optical conductivity is measured using the same type of sample heated by frequency-doubled Ti:Sapphire laser pulses (400nm) of 45 fs pulse width. In this case, laser absorption occurs through the excitation of 5d electrons into the conduction band. For this comparison, we choose data at 0.54 ps after laser excitation when the heated electrons are

thermalized, resulting in the same temperature and density conditions within the uncertainties as probed by our THz experiments.

The data reported in Ref. 13 were obtained with 300 nm-thick gold foil samples heated by 800 nm, 30 fs-long laser pulses, and the measurements were taken at 0.1ps after onset of laser heating. There data show significantly higher electron scattering frequencies as a function of T_e compared to our data. We believe that these differences can be explained by the following:

- (1) The data of Ref. 13 may have probed conditions with energy density gradients. This may arise due to the specific energy absorption mechanism of 800nm laser pulses where the leading part of the laser pulse deposits more energy near the sample surface, resulting in higher electron collision frequencies in that region and that can absorb significant energy in the trailing part of the pulse. This energy absorption mechanism is described in [S. Kirkwood, et al., Phys. Rev. B.79, 144120 (2009)]. On the other hand, our measurements are carried out in samples with very small gradients, as described in the supplemental information.
- (2) The use of fairly thick targets prevents the use of TTM to determine the temperatures. In Ref. 13, the electron temperature is estimated by comparing the estimated conduction electron density with the occupation electron density of states from DFT calculation. Here, they classify conduction electrons as those with energies above the Fermi energy/chemical potential. This method only provides a rough estimate and neglect $5d$ and $6s/p$ orbital hybridization.

To address these points, we have added the following sentences to the first paragraph of the section “Time and temperature dependence of electron scattering frequencies”:

The significantly higher v_e data reported in Ref. (13) can be explained by the following: (1) this study uses 800 nm, 30 fs laser pulses to excite 300 nm-thick samples, which could lead to larger longitudinal energy density gradients at the time of measurement at 0.1ps after laser excitation; (2) the use of fairly thick targets prevents the use of TTM to determine the temperature and their estimations of T_e using the occupation electron density of states are complicated by effects of electron orbital hybridization (16).

It seems strange to me that the author choose to present time evolution of electrical conductivity in the supplemental (Figure S6) as this is an important result for the community. It should be part of the paper. Time evolution of DC conductivity allows for comparison with time dependent models. For example, this is a way to test electron-phonon coupling which is another problematic quantity for WDM.

Response to the above comment:

We comply with the reviewer and added the requested information in the main text (resulting in a new Fig. 4). This figure and the corresponding caption are adopted from Fig. S6 of the supplemental information. We also added a new section “Temporal evolution of the electrical conductivity” to discuss these measurements:

Temporal evolution of the electrical conductivity

By varying the delay between the XUV-FEL pump and the THz probe pulse train allows us to measure the temporal evolution of the electrical conductivity over a 35 ps-long time window, spanning the transition from the solid state to a laser-excited solid and into the warm dense matter (plasma) state. Figure 4 shows the electrical conductivity as a function of time for an energy density of 0.89 ± 0.18 MJ/kg and assuming constant sample thickness for the whole time interval. Following the abrupt decrease right after $t = 0$, the conductivity continues to decline at a slower rate. It follows an exponential trend until about $t = 15$ ps as indicated by the linear dependence on the semi-log plot, cf. Fig. 4. However, at $t = 14.5$ ps we observe a discontinuity determined by a two-segment exponential fit; this time coincides with the melting phase transition and the onset of hydrodynamic expansion (8, 9). The sample expansion at $t > 15$ ps is likely accompanied by an increased sample thickness d as well as the development of density gradients. Future work will combine these data with independent measurements of the density profile to determine the transition to the classical plasma regime. These observations show that the multi-cycle THz probe determines σ accurately for strongly excited solids and provides information on phase transitions. In the following section, we focus on the data up to $t = 15$ ps since the target thickness is well constrained until melting and, in addition, independently measured accurate structure factors exist for these conditions.

Fig. 4: **Time-resolved electrical conductivity σ_0 of gold (logarithmic vertical axis) at mean excitation energy density of 0.89 ± 0.18 MJ/kg measured by multi-cycle THz-FEL at various time delays. Each section covers a time interval of 5 ps and the data presents an average over 4-6 shots. The error-bars include the standard deviation and the systematic error of the measurements. The data after 0 ps are fit by a two-segment exponential decay function represented by the dashed and the dotted lines. The intersection of these two curves at $t = 14.5$ ps coincides with the melting phase transition of gold.**

Accordingly, in the Supplemental Information, we have deleted the following paragraph in the section of “Multi-cycle THz data analysis”:

Figure S6 shows the time-resolved electrical conductivity ... This observation shows that the multicycle THz determines σ accurately for strongly excited solids and is sensitive to target expansion and melting.

We also modify the following sentences in the subsequent paragraph to keep it in the correct context. We changed the supplemental information text from:

For comparison, the results of the frequency-domain analysis using single-bend-emitted THz radiation is also presented. The latter is generated as the electron bunches are deflected by the dumping magnet located after the undulators ...

to

Fig. S7 shows the comparison of the time resolved electrical conductivity determined from our analysis with the more commonly applied THz time-domain spectroscopy (THz-TDS) analysis. The latter uses single-cycle THz pulse that is generated when the electron bunches are deflected by the dumping magnet located after the undulators ...

Please note that the original Fig. S6 becomes Fig. S7 in the updated version.

I am less convinced by the theoretical interpretation of the results.

I don't understand why the authors introduce a Drude-Smith model? Following ref. 27, this formulation could be necessary near the percolation regime i.e. for film thickness around 7nm. With a 30 nm film, there is no need to introduce this model. At the end, they determine a C value -0.17 ± 0.16 . Considering the error bars, the C value is nearly 0 as expected from 27 and the two curves (Drude and Drude-Smith are identical).

The author should justify why they think that they need this Drude-Smith model for their experimental condition. In my opinion, the use of Drude-Smith model here is not relevant. There is a typo in the reference for Drude-Smith model, it should be PRB 64 155106 (2001).

Response to the above comment:

We comply with the reviewer and move all reference to the Drude-Smith model into the supplemental information. In the main text and figures we use the Drude model to describe the broadband electrical conductivity.

We also thank the reviewer to mention Ref. 27, and that thin gold film samples deposited on silicon wafers do not show percolation effects as long as their thickness is above 10 nm. We move the comparison with the Drude-Smith model to the supplemental information. This comparison is valuable because we used 30nm free-standing samples that were deposited on NaCl crystals and then lift-off in water. As the reviewer points out these manufacturing differences are not important and the SI may be a suitable way to document this fact.

To comply with the reviewer, the following changes are made:

- 1) We moved the description and the figures related to the Drude-Smith model into the supplemental information
- 2) All figures in the main text have been updated and use only the Drude model.

In addition, the following modifications are applied to the manuscript:

- (1) In the section “Comparison of DC electrical conductivity and optical conductivity data”, the following sentences:

Further, corrections for carrier backscattering in nano films can be taken into account by the Drude-Smith model, ... The latter shows significantly lower DC conductivities σ_0 than obtained by our THz measurements,

are now modified to:

In Figure 3 we applied the Drude model to relate $\sigma(\omega)$ between the THz and the optical regimes,

$$\sigma(\omega) = \frac{n_e e^2 \tau}{m(1 - i\omega\tau)}$$

where n_e is the carrier electron density, e is the electron charge, $m = 9.1 \times 10^{-31}$ kg is the electron effective mass in gold (32), $\tau = 1/\nu_e$ is the electron scattering time, which is the inverse of the total electron momentum scattering frequency ν_e . We take the real part of Eq. 2 to fit the data in Fig. 3, i.e. $\sigma_r(\omega) = \frac{n_e e^2 \tau}{m(1 + \omega^2 \tau^2)}$. At $T_e = 300$ K, the electron density results in $n_e = 5.9 \times 10^{28} \text{ m}^{-3}$, corresponding to one carrier electron per atom (33). The electron scattering time is determined using the conductivity data measured in the three regimes by 1) the four-point probe at zero frequency, 2) the THz-FEL probe at 2.8 THz peak frequency, and 3) the optical probe laser at 1.55 eV. At elevated temperatures, only the THz and optical conductivity data are used to solve for n_e and t in the Drude formula. Figure 3 shows that the Drude model can satisfactorily describe the frequency dependency of the electrical conductivity from the AC to the DC regime if data are available in both regimes. The success of the Drude model was indeed expected as long as the foil thickness is above 10 nm and percolation can be neglected (27 and the Supplemental Information). Further, we find that the frequency response of $\sigma_r(\omega)$ is effectively constant for photon energies from zero to 0.012 eV, resulting in $\sigma_r(\omega) \approx \sigma_0$ in this regime, i.e., the DC conductivity value is identical to the THz conductivity. The dashed curves in Fig. 3 indicate the results when the same model is applied using only the optical data at 1.55 eV to

extrapolate to the DC conductivity regime. It leads to significantly lower DC conductivities σ_0 than obtained by our THz measurements...

(2) In the section “Time and temperature dependence of electron scattering frequencies”, the following sentences:

Applying the Drude or Drude-Smith models (Fig. 3) to the time-dependent THz data determines the total electron scattering frequency ν_e . The data from both models show a similar rapid increase right after the XUV-FEL excitation, ...

are changed to:

Applying the Drude model (Fig. 3) to the time-dependent THz data determines the total electron scattering frequency ν_e . The data show a rapid increase right after the XUV-FEL excitation, ...

(3) In Figure 3, the curves of the Drude-Smith model fit are removed. The updated figure and corresponding capture are shown below:

Figure 3: The electrical conductivity as a function of photon energy for various excitation conditions of solid gold. The broadband electrical conductivity data (real part, σ_r) at electron temperatures of $T_e = 300$ K (blue), 9000 ± 900 K (green) and 16000 ± 1500 K (red) are indicated by the different colors; the THz conductivity at small photon energies (0.75-3.75 THz in frequency, or 3-15.5 meV in photon energy) are shown as square blocks that cover the range of frequency components in the THz cycles (horizontal dimension) and **uncertainties of the inferred conductivity values (vertical dimension), the optical conductivity at higher photon **energies** (1.55 eV) from Ref. (30) are presented as diamonds, and the DC conductivity at room temperature measured with four-point probe is pointed out by a black arrow. **The solid curves represent the Drude model fit through the high and low frequency data, and the dashed curves represent Drude model extrapolations from the optical conductivity values at 1.55 eV. The conductivity measured with THz radiation agrees well with the DC conductivity from four-point probe measurements, while the extrapolation from optical measurements shows significant discrepancies.****

(4) Fig. 5 (previously Fig. 4), uses the results obtained from the Drude model. The figure is shown above on page 4 of this reply. The caption of the figure is also changed from:

Electron scattering frequencies as functions of time and temperatures determined from broadband conductivity fitting by the Drude-Smith model.

To:

Electron scattering frequencies as functions of time and temperatures determined from a broadband conductivity fit **using the Drude model.**

In the Supplemental Information, the following changes are made:

A new section “**Comparing the Drude model and the Drude-Smith model fit to the Broadband conductivity data**” is added, and the content in this section:

To confirm that the Drude model is sufficient to describe the frequency dependent electrical conductivity of our samples, we also use the Drude-Smith model to fit the broadband conductivity. The Drude-Smith model has similar format as the Drude model:

$$\sigma(\omega) = \frac{n_e e^2 \tau}{m(1 - i\omega\tau)} \left[1 + \frac{C}{1 - i\omega\tau} \right]$$

where n_e is the carrier electron density, e is the electron charge, m is the electron effective mass, $\tau = 1/\nu_e$ is the electron scattering time, which is the inverse of the total electron momentum scattering frequency ν_e , and the parameter C , ranging between 0 and -1, is introduced to account for electron back scattering from nano-structures in the sample (22, 23). It returns to the Drude model when $C=0$. We take the real part of Eq. 3 to fit the broadband conductivity data, i.e. $\sigma_r(\omega) = \frac{n_e e^2 \tau}{m(1+\omega^2\tau^2)} \left[1 + \frac{C(1-\omega^2\tau^2)}{1+\omega^2\tau^2} \right]$. Here, we first determine C at room temperature with $n_e = 5.9 \times 10^{28} \text{ m}^{-3}$, corresponding to one carrier electron per atom (24), and the electron effective mass $m = 9.1 \times 10^{-31} \text{ kg}$ in gold that is equal to its rest mass (25). Including the THz data, the optical data at 1.55 eV, and the four point

probe data, we calculate $C = -0.17 \pm 0.16$ for our thin foil samples, where the error bar is determined from the fit through the boundaries of the experimental error. Better determination of the C parameter requires more careful measurements of the DC, THz and optical continuities on the same samples, or even include additional measurements in the infrared regime. We find that the frequency response of $\sigma_r(\omega)$ is effectively constant for photon energies less than 0.012 eV, resulting in $\sigma_r \approx \sigma_0$ in this regime. In Fig. S9, the solid and dotted curves show the frequency-dependent conductivity values fit by the broadband Drude and Drude-Smith models respectively. These two curves are almost identical, indicating that the electron back scattering does not play a significant role in our samples, and the broadband electrical conductivity is adequately described by the Drude model.

In the reference list, we have corrected the page number from **115106** to **155106** in the citation of the Drude-Smith model (now only shown in the Supplemental Information). We appreciate the reviewer spotted this typo.

Concerning electron scattering frequency, they use simple model and many assumptions to deduce ee and ei scattering frequencies and I am not sure that the conclusions drawn are precise enough to write at the end of the abstract that they "elucidate the microscopic origin of electrical resistivity".

Response to the above comment:

The reviewer is correctly pointing out the model assumptions used in deducing e-e and e-i scattering frequencies. We agree with the reviewer that a less glorious and more specific statement would be better. We changed the sentence from:

Our results allow the direct determination of the electron-electron and electron-ion scattering frequencies that elucidate the microscopic origin of the electrical resistivity.

to

Our results allow the direct determination of the electron-electron and electron-ion scattering frequencies **that are the major contributors to the electrical resistivity.**

Reviewer #2 (Remarks to the Author):

Dear Editor,

I have reviewed the paper by Z. Chen " Ultrafast Multi-cycle Terahertz measurements of the electrical conductivity in strongly excited solids". The reported experiment is using a pump-probe approach: a 13.5 nm pump at intensity $>2 \text{ TW/cm}^2$ and a 100-300 μm relatively broadband probe to measure time dynamics of ionization and heating/melting of a golden nanofoil. Because the wavelength of the probe is much larger than the sample thickness, they relate the measured amplitude of the transmitted probe first to foil conductivity (before the pump pulse arrived) and then to the plasma conductivity. The former is calibrated against a true direct measurements of DC conductivity of the foil on the absolute scale. The deduced plasma conductivity is correlated to the initial foil conductivity. Overall, the experimental techniques, especially in part of time resolved single-shot electro-optic sampling for THz analysis, are impressive. The results obtained are significant and interesting for the laser-fusion and high-energy density physics communities. Combination of novel parameter space (relatively cold $\sim 1 \text{ eV}$ XUV driven plasma) and the results qualify this paper for publication in Nature Communications. However, several specific issues presented below are ought to be addressed prior to publication.

We appreciate the comments by Reviewer #2. We have complied with all requests and answered all questions and comments in the detailed point-by-point responses below. Thank you for your time and assessment.

1) Throughout the paper the authors make impression that they present direct measurements of DC electrical conductivity. Without any doubt data on THz transmission correlate with electrical conductivity, as was shown e.g.in Ref. 25 for THz probe reflection. But it needs to be clear stated that these are not direct measurements of DC electrical conductivity.

Response to the above comment:

We agree that we have measured the transmission of radiation in the THz regime, from which we can infer the DC conductivity of our samples both at room temperature and under heated conditions. To make this clear in our manuscript, we have complied with the reviewer and made the following corrections:

(1) The first sentence in the abstract is changed from

Key insights in materials at extreme temperatures and pressures can be gained by **direct measurements of the electrical conductivity.**

to

Key insights in materials at extreme temperatures and pressures can be gained by **novel measurements that determine the electrical conductivity.**

(2) Also in the abstract:

Our results allow the **direct determination**

Is changed to:

Our results allow the determination

(3) Added the following sentence in the section of “Conclusions and Outlook”

We have shown that in the warm dense matter regime the DC conductivity is well approximated by the electrical conductivity at THz frequencies.

2) In several places the authors call the ionization by 13.5 nm radiation as XUV excitation. Typically excitation process is the process that takes place below the ionization threshold and quantum structure of atoms plays a role. Peak Intensities of the pump in the study around 2TW/cm² are much above the typical ablation threshold of GW/cm², so plasma is generated. Indeed the plasma is non-Maxwellian and far from equilibrium but it is still a plasma and this is ought to be clarified.

Response to the above comment:

We appreciate that the reviewer is pointing out these facts.

The XUV-FEL first ionizes *5d* and *5p* electrons [J. J. Yeh et al., At. Data Nucl. Data Tables 32, 1 (1985)] in our gold foils initially. The ionized electrons quickly recombine by Auger decay and collision recombination. After some cascade processes, the electron system is thermalized in ~100fs time scale [N. Medvedev et al., Phys. Rev. Lett.107, 165003(2011)] to a temperature of 16000K, or 1.5eV according to the electron specific heat. This temperature is only a fraction of the Fermi energy of gold (5.5eV). Therefore, the electrons should follow the Fermi distribution. These non-ideal electron plasmas are degenerate and strongly coupled, and they are more generally referred to as the warm dense matter. In summary, the XUV-FEL ionizes the samples initially, but by the time we measure the THz conductivity at $t > 0.7\text{ps}$, the samples may be described as an excitation state. This is due to the fact that significant thermalization will have occurred at these time delays

To comply with the reviewer, we clarified these processes in the manuscript and supplemental information and made the following modifications:

In the manuscript:

(1) The second sentence in the abstract is changed from

Free-electron laser pulses can **excite matters out of equilibrium on femtosecond time scales, ...**

to

Free-electron laser pulses can **ionize and excite** matters out of equilibrium on femtosecond time scales, ...

(2) The following sentence in the first paragraph in the section “Electrical conductivity from THz measurements” is changed from

The 150 fs XUV-FEL radiation at a wavelength of $\lambda=13.6$ nm with up to 160 μJ pulse energy was used to excite 30 nm-thick free-standing gold foils to a maximum electron temperature of $T_e=16,000$ K determined from the absorbed XUV energy density (0.9 MJ/kg).

to

The 150 fs XUV-FEL radiation at a wavelength of $\lambda=13.6$ nm and with up to 160 μJ pulse energy was used to ionize the 5d and 5p electrons in 30 nm-thick free-standing gold foils. Subsequently, the sample thermalizes into a degenerate plasma state reaching maximum electron temperatures of $T_e=16,800$ K, i.e., 1.5 eV at peak intensities of $5\times 10^{11}\text{W}/\text{cm}^2$.

In the supplemental information, the following sentences in the section of “XUV-FEL pulse energy deposition and temperature estimation” are changed from

The energy absorption is dominated by bound-free transitions of the 5d and 5p electrons (2), and the absorption ratio of XUV energy is expected to be almost constant in our experimental condition. This is due to the fact that the amount of photo ionization induced by 91.2 eV electrons is only 0.02 electron per atom at the maximum excitation energy density of 1MJ/kg.

to

The energy absorption is dominated by **the ionization** of the 5d and 5p electrons (2), and the absorption ratio of XUV energy is expected to be almost constant in our experimental condition. **Taking into account the efficiency of the XUV optics, at maximum pulse energy of 160 μJ , about 75 μJ is focused on the sample surface, resulting in a maximum intensity of $5\times 10^{11}\text{W}/\text{cm}^2$. The maximum absorbed energy density is 1 MJ/kg, corresponding to a photon ionization rate of 0.02 eV using the 91.2 eV XUV photons. The ionized electrons are expected to be thermalized in 100 fs time scale (3), leading to a maximum electron temperature of 16,800 K, or 1.5 eV according to the two-temperature model calculations. Because this temperature is only a fraction of the Fermi energy (5.5 eV in gold), the electrons are expected to follow the Fermi distribution.**

3) Following 2). It will be useful for a reader to know that T_e is ~ 1 eV and the XUV intensity $\sim 2\text{TW}/\text{cm}^2$ to be conveniently connected with other laser-plasma studies.

Response to the above comment:

We comply with this suggestion and we added the requested information above in our response to the reviewers' 2nd comment.

In the manuscript we mention $5 \times 10^{11} \text{ W/cm}^2$ as the maximum XUV intensity. It takes in account the efficiencies of the XUV optics, and the enclosed energy in the FWHM focal spot. This information is now specified in the "XUV-FEL Pump Pulse" section in *Materials and Methods*. The following sentence is added:

Half of the FEL energy is enclosed in the FWHM area, resulting in the maximum intensity of $5 \times 10^{11} \text{ W/cm}^2$ on target.

4) At the very core of the paper is comparison of measured and deduced from measurements electrical conductivity of Au with multiple theoretical models including a common Drude, Drude-Smith, etc models. But there is a lack of comparison with data obtained by other groups, e.g from Ref. 25 on Al nanofoils, etc.

Response to the above comment:

We agree with the reviewer that it is helpful to compare our determined electrical conductivity data with data reported by other groups/other studies. To comply with the reviewer, now Fig. S13 in the supplementary information shows the comparison with the extrapolated DC conductivity from all other studies at similar heated sample conditions [Z. Chen et al., Phys. Rev.Lett.110, 135001 (2013), A. Ng et al., Phys. Rev. E94, 033213 (2016), K. Widmann et al., Phys. Rev. Lett.92, 125002 (2004), and C. Fourment et al., Phys. Rev. B89,116110(R) (2014)]. Regarding the THz conductivity measurements of Aluminum [K.Y. Kim et al., Phys. Rev. Lett. 100, 135002 (2008)], we prefer to reference it in the manuscript instead of comparing the data directly, because these data are taking on a difference element.

In the supplementary information, we also modified the title of the section "Extrapolating the DC conductivity from measured optical conductivity" to "**Extrapolating the DC conductivity from measured optical conductivity and comparison with the reported experimental data**". We also added the following discussions in this section to describe the comparison of different conductivity data:

For comparison, we also include other DC conductivity of isochorically heated gold extrapolated from optical conductivity data measured with 800nm lasers (28–30). In (28), a modified Drude model accounting for the non-Drude components in the imaginary part of the optical conductivity is applied to extrapolate DC conductivity from the optical conductivity reported in (13). In (29), the optical conductivity at 2 - 3ps after laser heating is used, where higher ionic temperature than this study is expected, and the authors use Eq. 4 to extrapolate DC conductivity. In (30), only the electron density n_e and the electron scattering frequency ν_e are reported, and we use them to construct the DC conductivity data following the original Drude formula. We can see that all these extrapolated DC conductivity data are lower than our results measured by the THz-FEL consistent with the extrapolations shown in Figure 3 of our main manuscript.

Fig. S13 **Comparison of the DC conductivity (σ_0) of isochorically heated gold as a function of electron temperature (T_e) determined from our THz transmission measurements (red solid circles), and the extrapolation results from the optical conductivity of (13) (grey open squares) and (28) (open blue circles) using the original Drude model and a modified Drude model (27) (orange open diamonds). Also shown is the DC conductivity constructed by the Drude model using electron density n_e and scattering frequency ν_e reported in (29) (purple open triangles).**

5) The authors are using an XUV FEL laser operating in the SASE regime which is famous for a random spike structure in the time domain. These are rather difficult to account for. How high intensity spikes may influence their measurements and how it was included in analysis of the total experimental uncertainty?

Response to the above comment:

The reviewer is making a good point. Indeed, the temporal spikes are difficult to characterize. Few studies are found in the literature. The paper of G. Geloni, et al., New J. Phys. 12, 035021(2010) simulates the temporal spikes measured from the European XFEL, and their results are shown in Fig. R1. We can see that the peak intensity of the SASE spike is 4 times higher than the average value.

In general, as long as the XUV-FEL pulse is short compared to the delay until we perform the THz conductivity measurements (at $t = 0.7$ ps), the temporal profile of the XUV pulse should

have little effect on our results. The possible influence of this high intensity spikes on our measurements is whether they can change the absorption of our XUV-FEL pulses in our target by intensity-dependent nonlinear absorption. Earlier studies on aluminum have found the onset of nonlinear saturable absorption at XUV-FEL intensity above $10^{14}\text{W}/\text{cm}^2$ [B. Nagler et al., Nat. Phys.5, 693 (2009)]. In our experimental conditions, the average peak intensity is only $5 \times 10^{11}\text{W}/\text{cm}^2$, and the photo-ionization rate is 0.02 electrons per atom. Therefore, we don't expect any nonlinear absorption to occur in our experiment.

To address this point, we have added the following sentences to the 1st paragraph of the section "XUV-FEL pulse energy deposition and temperature estimation" in the supplemental information:

It is also worth to mention that XUV-FEL pulses generated from the self-amplified spontaneous emission (SASE) usually contain multiple random spikes whose peak intensities can be several times higher than the average intensity (4). In our experiment, they are not expected to cause any change to the energy absorption. This is consistent with previous measurements of aluminum (5) that have shown linear absorption for intensities up to $10^{14}\text{W}/\text{cm}^2$.

Fig. R1 Simulated temporal profile of SASE emission at wavelength of 0.1nm at the European XFEL [copied from G. Geloni, et al., New J. Phys.12, 035021(2010)]

6) In Figure 2a, transmission on-shot and post-shot are presented on the same time scale from -2 to 3 ps It is confusing to see postshot data before the shot arrived. Please elaborate. Also it will be useful to make an inset with a zoom-in image of a raw data. Now it is very difficult to digest.

Response to the above comment:

Thanks for indicating the confusions in the time axis of the post-shot data. We would like to make it clear that there are no XUV-FEL heating pulses that arrived on target for both the “pre-shot” and “post-shot” measurements. We use the same -2 to 3ps time axis on all these data to overlap them with the “on-shot” measurement cycle by cycle. The XUV-FEL pulses only arrives at the samples during the “on-shot” measurements at 0 ps. To make it clear, we have added the follow sentences to the caption of Fig. 2:

In the on-shot measurement, the XUV-FEL heating pulse arrives at 0 ps. No XUV-FEL pulses are present during the pre-shot and post-shot measurements.

We also agree to improve the visibility of the THz data traces shown in Fig. 2 (a). Although there is little space in the figure to show an inset, we added the magnified THz traces (both before and after balance detection) in Fig. S4 in the supplemental information. This figure is showed below with the corresponding caption.

We also added a paragraph in the section of “Multi-cycle THz data analysis” in the supplemental information to introduce this new figure:

The THz signal is measured by the single-shot electro-optics sampling method with the implementation of balanced detection (17). Fig. S4 shows examples of the balanced detection that correspond to the data presented in Fig. 2 of the main text. We can see that subtracting the signal recorded by one polarization state of the sampling laser (left signal) by another one recorded by the opposite polarization state (right signal) results in significant improvements to the signal to noise ratio of the data.

Fig. S4 Example data of the THz electric field acquired by the balanced detection for (a) pre-shot, (b) on-shot and (c) post-shot measurements that correspond to the data shown in Fig. 2 of the main text. The XUV-FEL heating pulse arrives at 0ps during the on-shot measurement. No XUV-FEL pulse is applied during the pre-shot and post-shot measurements.

In the main text, we also added the following sentence at the end of the “THz-FEL Probe pulse” section in *Material and Methods* to point to this new figure in the supplemental information:

Examples of the acquired data using this balanced detection method are shown in the Supplemental Information.

Reviewer #3 (Remarks to the Author):

Overall the article is well written and likely to be of interest to different communities.

We appreciate the comments by Reviewer #3. We have complied with all requests and answered all questions and comments in the detailed point-by-point responses below. Thank you for your time and for your assessment.

One issue is with the use of the Drude-Smith model over the Drude model. Based on the fitting the authors obtain a value for C of -0.17 ± 0.16 implying that the Drude model only sufficiently fits the data based on the error. While an alternative value of C of -0.17 may be a better fit, this also complicates the model. Additionally this error bound represents a relatively large portion of the allowed values for C. The authors should comment on the error bars and ways to reduce the error in the future either with more detailed measurements, using data from other experiments, etc.

Response to the above comment:

We agree with the reviewer that an extensive discussion of the Drude-Smith model in the main text is a distraction. We consequently moved all reference to this model to the supplemental information and focus the main text on the Drude model which is well established in this community. The discussions on the error bar of C, and the ways to reduce the error are included in the supplemental information.

To comply with the reviewer, we made the following amendments in the section “Comparison of DC electrical conductivity and optical conductivity data”, (see also response to reviewer #1):

In Figure 3 we applied the Drude model to relate $\sigma(\omega)$ between the THz and the optical regimes,

$$\sigma(\omega) = \frac{n_e e^2 \tau}{m(1 - i\omega\tau)}$$

where n_e is the carrier electron density, e is the electron charge, $m = 9.1 \times 10^{-31}$ kg is the electron effective mass in gold (32), $\tau = 1/\nu_e$ is the electron scattering time, which is the inverse of the total electron momentum scattering frequency ν_e . We take the real part of Eq. 2 to fit the data in Fig. 3, i.e. $\sigma_r(\omega) = \frac{n_e e^2 \tau}{m(1 + \omega^2 \tau^2)}$. At $T_e = 300$ K, the electron density results in $n_e = 5.9 \times 10^{28} \text{ m}^{-3}$, corresponding to one carrier electron per atom (33). The electron scattering time is determined using the conductivity data measured in the three regimes by 1) the four-point probe at zero frequency, 2) the THz-FEL probe at 2.8 THz peak frequency, and 3) the optical probe laser at 1.55 eV. At elevated temperatures, only the THz and optical conductivity data are used to solve for n_e and t in the Drude formula. Figure 3 shows that the Drude model can satisfactorily describe the frequency dependency of the electrical conductivity from the AC to the DC regime if data are available in both regimes. The success of the Drude model was indeed expected as long as the foil thickness is above 10 nm and percolation can be neglected (27 and the Supplemental Information).

Further, we find that the frequency response of $\sigma_r(\omega)$ is effectively constant for photon energies from zero to 0.012 eV, resulting in $\sigma_r(\omega) \approx \sigma_0$ in this regime, i.e., the DC conductivity value is identical to the THz conductivity. The dashed curves in Fig. 3 indicate the results when the same model is applied using only the optical data at 1.55 eV to extrapolate to the DC conductivity regime. It leads to significantly lower DC conductivities σ_0 than obtained by our THz measurements...

The following sentences (in red) are added in the new section “Comparing the Drude model and the Drude-Smith model fit to the Broadband conductivity data” in the supplemental information:

Including the THz data, the optical data at 1.55 eV, and the four-point probe data, we calculate $C = -0.17 \pm 0.16$ for our thin foil samples, where the error bar is determined from the fit through the boundaries of the experimental error. Better determination of the C parameter requires more careful measurements of the DC, THz and optical conductivities on the same samples, or even include additional measurements in the infrared regime.

In figure 4, the authors should add the fits for the Drude model in addition to the Drude-Smith model to better match figure 3 and allow the reader to better understand the differences. Adding this information to 4a should be sufficient if adding it to b-d makes the figure difficult to read.

Response to the above comment:

We comply with the reviewer and added the comparison of the total electron scattering frequency of ν_e determined from the Drude model in Figure 5 (Figure 4 in the original manuscript). In addition, we provide a comparison with the Drude-Smith model in the supplemental information in Fig. S10(a).

The updated Figure 5 is showed below:

Fig. 5 (originally Figure 4 in the manuscript): **Electron scattering frequencies as functions of time and temperatures determined from a broadband conductivity fit using the Drude model.**

The updated Fig. S10 comparing the ν_e determined from the Drude and the Drude Smith model is shown below

Fig. S10 Electron scattering frequencies as functions of time and temperatures determined from a broadband conductivity fit by the Drude-Smith model. (a) shows the total electron scattering frequency (ν_e) as a function of T_e determined from this work (blue symbols), compared to the data reported from optical measurements (28, 30) (grey open triangles and diamonds) and the theoretical calculations from Ref. (28) (red curve). **The ν_e data determined from the Drude model (light blue) are also show...**

We can see that the difference of electron scattering frequency determined from both the Drude and the Drude-Smith model is only 15%, which is comparable to the experimental precision. Therefore, we accepted the suggestion by the Reviewer #1 and shown only the Drude model results in Fig. 5(b), (c) and (d). The original data from the Drude-Smith model is now deposited to the supplemental information as Fig. S10.

The authors should clarify the sentence (bottom of page 3) "Finally, our results demonstrate the shortfalls of a simple Drude extrapolation." to clarify extrapolation from optical frequencies.

Response to the above comment:

We agree with the reviewer that the original sentence can cause confusions. To comply with the reviewer, we have revised the sentence as follows:

Finally, our results demonstrate that the Drude model can successfully describe the broadband electrical conductivity when interpolating between the THz and optical data, and disclose the shortfalls when extrapolating using only optical data.

The authors should briefly mention the accuracy and error range of the two temperature model in the text.

Response to the above comment:

In the two-temperature model, most of the parameters are well tested. The major uncertainty comes from the electron-ion energy coupling factor g_{ei} . We obtain this parameter reported from recent experimental study of [M.Z. Mo, et al., Science 360, 1451 (2018)]. This result is supported by recent theoretical calculations of [N. Medvedev Phys. Rev. B 102, 064302 (2020)].

In the paper by Mo et al., the error bar of g_{ei} is +/-20%. To comply with the reviewer and include this information in the manuscript, we perform the new TTM calculations using the upper and lower bound of the g_{ei} . The result is now included in Fig. S3 (b) in the supplementary information. The updated figure and caption are shown below as Fig. S6.

In addition, the following sentence is added in the last paragraph of the section “XUV-FEL pulse energy deposition and temperature estimation” in the supplementary information:

In the TTM calculations, we also include the 20% error-bar from the electron-ion energy coupling factor g_{ei} reported in (12), and the resulting uncertainties are shown in the pink and light-blue area.

To consider this error range in the deduced data, we include the ion temperature uncertainty from the TTM calculations in Fig. 5(d) of the manuscript (already shown in Fig. R5 above), and the description is in the last sentence of the caption:

and the error bars in temperature consider both the precisions of experimental measurements and the TTM calculations.

Fig. S6 Two temperature model calculated (a) electron temperature T_e and ion temperature T_i as functions of absorbed energy density at $t=0.7\text{ps}$ after the XUV-FEL excitation, and (b) temporal evolution of T_e and ion T_i at an absorbed energy density of 0.89 MJ/kg . **The pink and the light blue shadows represent the error estimates of the TTM calculations.**

Reviewers' Comments:

Reviewer #1:

Remarks to the Author:

I am completely satisfied with the new version of the manuscript. The authors have answered all my questions. The manuscript is now suitable for publication.

Reviewer #2:

Remarks to the Author:

Dear Editor,

I have reviewed the revised manuscript. It is an interesting paper and all my concerns are addressed satisfactory. It can be published as is.

Reviewer #3:

Remarks to the Author:

The authors have sufficiently addressed my issues and is likely to be of interest to a number of communities.

Point-by-point response to reviewers' comments

Reviewer #1 (Remarks to the Author):

I am completely satisfied with the new version of the manuscript. The authors have answered all my questions. The manuscript is now suitable for publication.

We are pleased that Reviewer #1 is satisfied with our revision. We thank him/her for improving our manuscript.

Reviewer #2 (Remarks to the Author):

Dear Editor,

I have reviewed the revised manuscript. It is an interesting paper and all my concerns are addressed satisfactory. It can be published as is.

We are glad that all the concerns of Reviewer #2 have been addressed. We thank him/her for improving our manuscript.

Reviewer #3 (Remarks to the Author):

The authors have sufficiently addressed my issues and is likely to be of interest to a number of communities.

We feel delighted that all issues from Reviewer #3 are resolved. We thank him/her for improving our manuscript.